# Non-Linear Optical Properties of Biexciton in Ellipsoidal Quantum Dot

**DOI:** 10.3390/nano12091412

**Published:** 2022-04-20

**Authors:** Yuri Y. Bleyan, Paytsar A. Mantashyan, Eduard M. Kazaryan, Hayk A. Sarkisyan, Gianluca Accorsi, Sotirios Baskoutas, David B. Hayrapetyan

**Affiliations:** 1Department of General Physics and Quantum Nanostructures, Russian-Armenian University, Yerevan 0051, Armenia; yuri.bleyan@rau.am (Y.Y.B.); paytsar.mantashyan@rau.am (P.A.M.); eduard.ghazaryan@rau.am (E.M.K.); hayk.sarkisyan@rau.am (H.A.S.); 2Photonics Laboratory, Institute for Physical Research of NAS RA, Ashtarak 0203, Armenia; 3Institute of Electronics and Telecommunications, Peter the Great St. Petersburg Polytechnic University, 195251 Saint Petersburg, Russia; 4CNR NANOTEC, Institute of Nanotechnology, University Campus Ecotekne, 73100 Lecce, Italy; gianluca.accorsi@nanotec.cnr.it; 5Department of Materials Science, University of Patras, 265 04 Patras, Greece; bask@upatras.gr

**Keywords:** oblate ellipsoidal quantum dot, exciton, biexciton, third-order susceptibility, two-photon absorption

## Abstract

We have presented a theoretical investigation of exciton and biexciton states for the ground and excited levels in a strongly oblate ellipsoidal quantum dot made from GaAs. The variational trial wave functions for the ground and excited states of the exciton and biexciton are constructed on the base of one-particle wave functions. The energies for the ground and excited levels, depending on the ellipsoidal quantum dot’s geometrical parameters, are depicted in the framework of the variational method. The oscillator strength of the transition from exciton to biexciton states for ground and excited levels is investigated as a function of the ellipsoidal quantum dot’s small and large semiaxes. The third-order optical susceptibilities of ground and excited biexcitons around one-photon and two-photon resonances are calculated as a function of the photon energy. The dependences of third-order optical susceptibilities for the ground and excited levels on the photon energy for different values of the ellipsoidal quantum dot’s semiaxis are revealed. The absorption coefficients in the ellipsoidal quantum dot, both for ground and excited states of exciton and biexciton, are calculated. The absorption coefficients for the ground level of exciton and biexciton for the fixed value of the large semiaxis and for the different values of the small semiaxis are determined. Finally, the two-photon absorption coefficient of the biexciton in the GaAs ellipsoidal quantum dot is computed.

## 1. Introduction

Exciton (X) and biexciton (XX) states in semiconductor quantum dots (QD) have been investigated both theoretically and experimentally by different authors [1,2,3,4,5,6]. As it is well known, exciton represents electron-hole pair, while biexciton consists of two excitons, i.e., it is a four-particle system. These quasiparticles play an important role in different effects, in particular, in Rabi oscillations [7] and in the optical Stark effect [8]. On the other hand, the transformation of excitons to biexciton leads to another class of interesting physical phenomena. In this regard, it is worth to note a change in the nonlinear third-order susceptibility, two-photon absorption, the appearance the shift of photoluminescence emission peaks, etc. [9,10,11,12].

Numerous works are devoted to the investigation of lowest-state or ground properties of excitons and biexcitons [13,14,15,16,17,18,19]. For example, in [13] the authors have investigated, theoretically and experimentally, the excitonic absorption spectrum, and in [15] the authors have considered the impact of excess electron occupation on the exciton’s optical properties. Additionally, in [17] the hybrid nanostructure model has been discussed for the efficient creation of the biexciton state.

The optical nonlinearity associated with the transitions between the sublevels of the excitons is expected to be available for novel infrared devices. Therefore, it is very important to study the optical properties not only of the lowest state but also of the higher excited states of confined excitons or multi-excitons [20]. Additionally, the biexciton state is known to have a relatively higher oscillator strength for two-photon generation [11,21]. Consequently, the optical nonlinearity should be enhanced by the oscillator strength. On the other hand, authors in various works are focused on the non-linear properties of biexcitons [22,23,24,25,26]. In particular, in [24] the biexciton states have been discussed in the framework of the variation method. The optical nonlinearity via the exciton and biexciton states is considered, based on the three levels, and the third-order susceptibility has been calculated. In another work [25], linear as well as non-linear optical properties for intersubband electronic transitions, associated with a biexciton, have been investigated. By using computed energies and a wave function, the optical absorption coefficients and the refractive index changes have been calculated. One of the most important results is that the optical absorption saturation intensity can be tuned by the confinement potential frequency. To continue, in [26] the theoretical study of linear and non-linear properties has been presented, based on a three-level model. The refractive index changes around one-, two-, and three-photon resonance have been calculated. Moreover, the authors found the strongly dependence of QD’s size on the third-order refractive index changes for three photon process.

Novel technologies make the growth of QDs of different types and geometries possible. Numerous works are devoted to the investigation of the physical and optical properties of QDs with relatively simple and complicated geometry [27,28,29,30]. Ellipsoidal QD is one of the examples of complicated geometry. Such QD has wide application in cases where there is a need to control the energy in a wide range. This manipulation is due to the fact that ellipsoidal QD has two geometrical parameters—small and large semiaxes. It is also worth highlighting that Kohn theorem is implemented both theoretically and experimentally in ellipsoidal QDs [31,32]. Thus, the theoretical investigation of exciton and biexciton complexes in ground and excited states in ellipsoidal QD is a problem. Additionally, it should be noted that we will focus on a special type of the ellipsoidal QD, namely, on the strongly oblate one, which supposes the quasi two- dimensional character of the biexciton. This means that two electrons and two holes will be localized in the same plane. Thus, the properties of the exciton and biexciton will be marked as two-dimensional.

In this paper, we will theoretically investigate non-linear optical properties of biexcitons of ground and excited states in GaAs strongly oblate ellipsoidal QD (SOEQD). The paper is organized as follows: Section 2 is devoted to the exciton and biexciton states in ellipsoidal QD; non-linear optical properties are described in detail in Section 3; the discussion of the obtained results is presented in Section 4; and, finally, the short summary of the main results is included in the last section.

## 2. Exciton and Biexciton States

As it is mentioned above, we will consider a specific type of ellipsoidal QD, namely, SOEQD. The particle’s confining potential energy in cylindrical coordinates, localized in SOEQD with impenetrable walls, has the following form:(1)$$U_{conf}\left(\rho ,\varphi ,z\right)=\left\{ \begin{array}{l} 0,\;\;\frac{\rho^{2}}{a^{2}}+\frac{z^{2}}{c^{2}}\leq 1\\ \infty ,\;\;\frac{\rho^{2}}{a^{2}}+\frac{z^{2}}{c^{2}}>1 \end{array}\right.,\;\;\;a>>c$$

Here, c and a are the small and large semiaxes of the SOEQD, respectively. In the case where the finite confinement potential is considered, some new physical phenomena will appear, for example, the possibility of the leakage of the electron and the hole for the QD to the surrounding media. It is obvious that in this case the biexciton binding energy will be less and the biexciton stability can be broken, depending on the geometrical parameters of the ellipsoidal QD. However, we have considered the case where the QD is surrounded by the dielectric media, so that the confinement energy prevails over the energies, mentioned in the article. From a theoretical point of view, the problem becomes more complicated and requires special numerical methods to be implemented.

The Hamiltonians of the biexciton and exciton have the forms:(2)$$\begin{array}{l} {\hat{H}}_{XX}\left({\vec{r}}_{1},{\vec{r}}_{2},{\vec{r}}_{\alpha },{\vec{r}}_{\beta }\right)=\sum_{j}\frac{{\hat{P}}_{j}^{2}}{2m_{j}^{*}}+\sum_{j}U_{conf}\left({\vec{\rho }}_{j},z_{j}\right)+V_{\mathrm{int}}\left({\vec{r}}_{1},{\vec{r}}_{2},{\vec{r}}_{\alpha },{\vec{r}}_{\beta }\right),\\ {\hat{H}}_{X}\left({\vec{r}}_{1},{\vec{r}}_{\alpha }\right)=\sum_{i}\frac{{\hat{P}}_{i}^{2}}{2m_{i}^{*}}+\sum_{i}U_{conf}\left({\vec{\rho }}_{i},z_{i}\right)+V_{\mathrm{int}}\left({\vec{r}}_{1},{\vec{r}}_{\alpha }\right), \end{array}$$
where j={1,2,α,β}, i={1,α}, ${\vec{r}}_{1}$ and ${\vec{r}}_{2}$ are electrons’ coordinates, ${\vec{r}}_{\alpha }$ and ${\vec{r}}_{\beta }$—holes’ coordinates. Interparticle interaction energies are in Hamiltonians for biexciton and exciton $V_{\mathrm{int}}\left({\vec{r}}_{1},{\vec{r}}_{2},{\vec{r}}_{\alpha },{\vec{r}}_{\beta }\right)$ and $V_{\mathrm{int}}\left({\vec{r}}_{1},{\vec{r}}_{\alpha }\right)$, including interaction between electrons, between holes, and between electrons and holes. This interaction for biexciton and exciton problems has the following forms:(3)$$\begin{array}{l} V_{\mathrm{int}}\left({\vec{r}}_{1},{\vec{r}}_{2},{\vec{r}}_{\alpha },{\vec{r}}_{\beta }\right)=\frac{e^{2}}{\varepsilon \left|{\vec{r}}_{1}-{\vec{r}}_{2}\right|}+\frac{e^{2}}{\varepsilon \left|{\vec{r}}_{\alpha }-{\vec{r}}_{\beta }\right|}-\frac{e^{2}}{\varepsilon \left|{\vec{r}}_{1}-{\vec{r}}_{\alpha }\right|}-\frac{e^{2}}{\varepsilon \left|{\vec{r}}_{1}-{\vec{r}}_{\beta }\right|}-\frac{e^{2}}{\varepsilon \left|{\vec{r}}_{2}-{\vec{r}}_{\alpha }\right|}-\frac{e^{2}}{\varepsilon \left|{\vec{r}}_{2}-{\vec{r}}_{\beta }\right|},\\ V_{\mathrm{int}}\left({\vec{r}}_{1},{\vec{r}}_{\alpha }\right)=-\frac{e^{2}}{\varepsilon \left|{\vec{r}}_{1}-{\vec{r}}_{\alpha }\right|}. \end{array}$$

It is noteworthy that the specific geometry of the SOEQD makes possible the usage of important approximation for the nature of the Coulomb interaction. Based on the oblate geometry of the ellipsoidal QD and taking into account strong size quantization in axial direction, it can be shown that the problem (exciton and biexciton) has a strongly pronounced two-dimensional character (see detailed calculations in [6]). Using the justification, presented in [6], one can get to the following two-dimensional Schrödinger equation:(4)$$\left[\sum_{j}\left(\frac{{\hat{P}}_{xj}^{2}}{2m_{j}^{*}}+\frac{{\hat{P}}_{yj}^{2}}{2m_{j}^{*}}\right)+\sum_{j}\frac{m_{j}^{*}\Omega_{j}^{2}}{2}\rho_{j}^{2}+\;V_{\mathrm{int}}^{2D}\left({\vec{\rho }}_{1},{\vec{\rho }}_{2},{\vec{\rho }}_{\alpha },{\vec{\rho }}_{\beta }\right)\right]\Psi_{XX}\left({\vec{\rho }}_{1},{\vec{\rho }}_{2},{\vec{\rho }}_{\alpha },{\vec{\rho }}_{\beta }\right)=E_{XX}^{2D}\Psi_{XX}\left({\vec{\rho }}_{1},{\vec{\rho }}_{2},{\vec{\rho }}_{\alpha },{\vec{\rho }}_{\beta }\right)$$
where $E_{XX}^{2D}=E_{XX}-\sum_{j}\frac{\pi^{2}\hbar^{2}}{8m_{j}^{*}c^{2}}$ and $\Omega_{j}=\frac{\pi \hbar }{2m_{j}^{*}ac}$ is the frequency of the parabolic confining potential.

Thus, the problem is reduced to the solving of the two-dimensional Schrödinger equation. Analogous to Equation (4), for the exciton one can get:(5)$$\left[\sum_{i}\left(\frac{{\hat{P}}_{xi}^{2}}{2m_{i}^{*}}+\frac{{\hat{P}}_{yi}^{2}}{2m_{i}^{*}}\right)+\sum_{i}\frac{m_{i}^{*}\Omega_{i}^{2}}{2}\rho_{i}^{2}+\;V_{\mathrm{int}}^{2D}\left({\vec{\rho }}_{1},{\vec{\rho }}_{\alpha }\right)\right]\Psi_{X}\left({\vec{\rho }}_{1},{\vec{\rho }}_{\alpha }\right)=E_{X}^{2D}\Psi_{X}\left({\vec{\rho }}_{1},{\vec{\rho }}_{\alpha }\right)$$

Since the two- or four-particle problem is analytically unsolvable, as a first step, it is necessary to obtain exciton and biexciton wave functions and energy spectra as a function of the ellipsoidal QD’s small and large semiaxes by using the variational method. For the variational calculations of biexciton and exciton states in SOEQD for the ground and excited states, single-particle wave functions and energies, obtained in SOEQD with impermeable walls from Ref. [30], are used. It is important to indicate that the obtained results for the one-particle problem have been done in the scope of the adiabatic approximation and are in good agreement with the numerical method’s results. The use of adiabatic approximation of the considered system is justified by the two-dimensional character of the QD.

The wave function and energy for a single particle problem in SOEQD have the following forms, accordingly [30]:(6)     ψ(ρ,φ,z)=eimφ2π(c1−ρ2a2)−1/2sin(πn2c1−ρ2/a2z+πn2)××2me*Ωeℏnr!Γ(|m|+1)Γ3/2(|m|+1+nr)e−me*Ωe2ℏρ2(me*Ωeℏρ2)|m|2F11{−nr,|m|+1;me*Ωeℏρ2},
(7)$$E=\frac{\pi^{2}\hbar^{2}n^{2}}{8m_{e}^{*}c^{2}}+\frac{\pi \hbar^{2}n}{2m_{e}^{*}ac}\left(N+1\right),\;\;\;\;N=0,1,2,\ldots$$

Here, m, n, and $n_{r}$ are quantum numbers (QN) describing the system. In particular, m is a magnetic QN, n is the QN in axial direction, $n_{r}$ is QN in radial direction, and $N=2n_{r}+\left|m\right|$ is principal QN. Finally, F11{a,b;x} is a confluent hyper-geometric function of the first kind. Note that for the excited levels we will consider levels by exciting radial QN since the excitation in the axial direction has a more significant impact on the total energy than in the radial direction. Hence, the first excited levels of the system will be defined by the radial QN.

The problem of constructing the wave functions of few-particle quantum systems based on one-particle approximation is extremely important and is largely dictated by the formulation of the problem itself. As is known, one of the most well-known approximations is the construction of fermion wave functions using Slater determinants [33]. This approach takes into account the asymmetry of the many-particle wave function with respect to particle permutations, as well as the Pauli principle. In the relativistic case, the problem becomes even more complicated. Here, it is necessary to take into account quantum dynamic effects. Moreover, if we take into account the fact that the wave functions themselves become operators, the issue of gauge invariance of the wave functions themselves requires a separate detailed study. In this regard, the work [34] should be noted, where the authors proposed a method for constructing the wave function of a multielectron atom based on the optimal selection of one-electron representations by minimizing the contribution to the radiation width of the studied atom of the many-electron interaction, which depends on the choice of gauge. The system considered in this paper is described in the framework of the nonrelativistic approximation and in the absence of external fields. Therefore, the issue of calibration arises only when the operator of the incident perturbation is chosen. We have considered the radiation incident normal to the plane of the QD section. Here, the Coulomb gauge divA=0 was chosen.

By using the variational method, we will perform calculations for the ground and excited levels of biexciton and exciton. For both cases, the variational functions are constructed on single-particle wave functions, which are correlated with each other by exponential function(s), containing variational parameters. These variational wave functions have the following form [11]:(8)$$\begin{array}{l} \Psi_{XX}\left({\vec{\rho }}_{1},{\vec{\rho }}_{2},{\vec{\rho }}_{\alpha },{\vec{\rho }}_{\beta }\right)=C\psi_{100}\left({\vec{\rho }}_{1}\right)\psi_{100}\left({\vec{\rho }}_{2}\right)\psi_{100}\left({\vec{\rho }}_{\alpha }\right)\psi_{100}\left({\vec{\rho }}_{\beta }\right)\times \\ \;\;\;\;\;\;\;\;\;\;\;\;\;\;\;\;\;\;\;\;\;\;\;\;\;\;\;\;\;\;\;\;\;\;\times e^{-\gamma \rho_{\alpha \beta }}\left\{e^{-\lambda \left(\rho_{1\alpha }+\rho_{2\beta }\right)-\delta \left(\rho_{1\beta }+\rho_{2\alpha }\right)}+e^{-\lambda \left(\rho_{1\beta }+\rho_{2\alpha }\right)-\delta \left(\rho_{1\alpha }+\rho_{2\beta }\right)}\right\}\\ \Psi_{X}\left({\vec{\rho }}_{1},{\vec{\rho }}_{\alpha }\right)=C\psi_{100}\left({\vec{\rho }}_{1}\right)\psi_{100}\left({\vec{\rho }}_{\alpha }\right)e^{-\mu \rho_{1\alpha }}. \end{array}$$

Here, C—normalization constant, $\rho_{jk}=\left|{\vec{\rho }}_{j}-{\vec{\rho }}_{k}\right|,\;\;j,k=\left\{1,2,\alpha ,\beta \right\}$, λ, δ, γ, and μ are variational parameters, which are determined after minimizing the following integrals:(9)$$\begin{array}{l} E_{XX}=\langle \Psi_{XX}({\vec{r}}_{1},{\vec{r}}_{2},{\vec{r}}_{\alpha },{\vec{r}}_{\beta })\left|{\hat{H}}_{XX}\right|\Psi_{XX}({\vec{r}}_{1},{\vec{r}}_{2},{\vec{r}}_{\alpha },{\vec{r}}_{\beta })\rangle \\ E_{X}=\langle \Psi_{X}({\vec{r}}_{1},{\vec{r}}_{\alpha })\left|{\hat{H}}_{X}\right|\Psi_{X}({\vec{r}}_{1},{\vec{r}}_{\alpha })\rangle \end{array}$$

It should be emphasized that in the literature devoted to the biexciton’s studies, it is accepted to use such wave functions [11,35,36]. The considered trial wave function is anti-symmetric with respect to electrons and holes.

## 3. Non-Linear Properties

Since we have already obtained the wave functions and energy spectra for exciton and biexciton, as a next step, we will investigate the oscillator strength of transition from exciton to biexciton states and the oscillator strength of excitonic transition. These oscillator strengths can be given by:(10)$$\begin{array}{l} f_{be}=\frac{2}{m_{0}\hbar \omega_{be}}{\left|\langle XX\left|p\right|X\rangle \right|}^{2}\\ f_{eg}=\frac{2}{m_{0}\hbar \omega_{eg}}{\left|\langle X\left|p\right|0\rangle \right|}^{2}, \end{array}$$
where |XX〉 and |X〉 denote the biexciton and exciton states, respectively; $m_{0}$ is the mass of free-electron; p is the momentum operator; and $\hbar \omega_{be}$ and $\hbar \omega_{eg}$ are the transition energy from exciton to biexciton states and the exciton energy, respectively. For the convenience, we have used special denotations: b will correspond to the biexciton level, e is for the exciton level, and g is for the ground. Please note that in our paper, the term “ground” corresponds to the case where no particle exists in QD.

For the investigation of the non-linear properties of biexciton, we will be limited by the case of three-level model. For the three-level model, the general expression of the third-order nonlinear optical susceptibility can be calculated by [37]:(11)$$\begin{array}{l} \chi^{\left(3\right)}\left(2\omega_{1}-\omega_{2};-\omega_{1},-\omega_{1},\omega_{2}\right)=-\frac{i{\left|\mu_{eg}\right|}^{4}}{2}\frac{1}{i\left(\hbar \omega_{eg}-2\hbar \omega_{1}+\hbar \omega_{2}\right)+\hbar \Gamma_{eg}}\frac{1}{i\left(\hbar \omega_{2}-\hbar \omega_{1}\right)+\hbar \Gamma_{e}}\times \\ \times \left(\frac{1}{i\left(\hbar \omega_{eg}-\hbar \omega_{1}\right)+\hbar \Gamma_{eg}}+\frac{1}{i\left(\hbar \omega_{2}-\hbar \omega_{eg}\right)+\hbar \Gamma_{eg}}\right)+\frac{i{\left|\mu_{eg}\right|}^{2}{\left|\mu_{be}\right|}^{2}}{4}\frac{1}{i\left(\hbar \omega_{be}-2\hbar \omega_{1}+\hbar \omega_{2}\right)+\hbar \Gamma_{be}}\frac{1}{i\left(\hbar \omega_{2}-\hbar \omega_{1}\right)+\hbar \Gamma_{e}}\times \\ \times \left(\frac{1}{i\left(\hbar \omega_{eg}-\hbar \omega_{1}\right)+\hbar \Gamma_{eg}}+\frac{1}{i\left(\hbar \omega_{2}-\hbar \omega_{eg}\right)+\hbar \Gamma_{eg}}\right)-\frac{i{\left|\mu_{eg}\right|}^{2}{\left|\mu_{be}\right|}^{2}}{4}\frac{1}{i\left(\hbar \omega_{eg}-2\hbar \omega_{1}+\hbar \omega_{2}\right)+\hbar \Gamma_{eg}}\frac{1}{i\left(\hbar \omega_{bg}-2\hbar \omega_{1}\right)+\hbar \Gamma_{bg}}\times \\ \times \left(\frac{1}{i\left(\hbar \omega_{eg}-\hbar \omega_{1}\right)+\hbar \Gamma_{eg}}\right)+\frac{i{\left|\mu_{eg}\right|}^{2}{\left|\mu_{be}\right|}^{2}}{4}\frac{1}{i\left(\hbar \omega_{be}-2\hbar \omega_{1}+\hbar \omega_{2}\right)+\hbar \Gamma_{be}}\frac{1}{i\left(\hbar \omega_{bg}-2\hbar \omega_{1}\right)+\hbar \Gamma_{bg}}\times \\ \times \left(\frac{1}{i\left(\hbar \omega_{eg}-\hbar \omega_{1}\right)+\hbar \Gamma_{eg}}\right), \end{array}$$
where $\hbar \omega_{1}$ and $\hbar \omega_{2}$ are energies of the first and second photons; $\hbar \omega_{ij},$ $\mu_{ij}$, and $\hbar \Gamma_{ij}$ denote the energy difference, the transition dipole moment between i and j levels, and the dephasing rate of the transition dipole moment, respectively; and $\Gamma_{e}$ is the population decay rate of the exciton state, which is inversely proportional to the radiative lifetime, hence $\Gamma_{e}=\frac{2ne^{2}\omega_{eg}{}^{2}}{3m_{0}s^{3}}f_{eg}$, where $n_{ref}$ is the refractive index of the material and s is speed of light. Hereafter, we will consider the case where two photons have the same energy $\hbar \omega_{1}=\hbar \omega_{2}\equiv \hbar \omega$.

The transition dipole moment is related to the oscillator strength as:(12)$$\mu_{ij}^{2}=\frac{\hbar e^{2}}{2m_{0}\omega_{ij}}f_{ij}$$

From Equation (11), one can get expressions for one-photon resonance at $\omega =\omega_{eg}$ and $\omega =\omega_{be}$, as well as an expression around two-photon resonance. Namely, for one-photon resonance, correspondingly, e→g and b→e transitions, the following expressions take place:(13)$$\begin{array}{l} -\frac{i{\left|\mu_{eg}\right|}^{4}}{2\hbar \Gamma_{\left|\right|}^{e}}\frac{2\hbar \Gamma_{eg}}{i{\left(\hbar \omega -\hbar \omega_{eg}\right)}^{2}+{\left(\hbar \Gamma_{eg}\right)}^{2}}\frac{1}{i\left(\hbar \omega_{eg}-\hbar \omega \right)+\hbar \Gamma_{eg}}\;\;;\;\;\;\;\omega =\omega_{eg}\\ \frac{i{\left|\mu_{eg}\right|}^{2}{\left|\mu_{be}\right|}^{2}}{4\hbar \Gamma_{\left|\right|}^{e}}\frac{2\hbar \Gamma_{eg}}{{\left(\hbar \omega -\hbar \omega_{eg}\right)}^{2}+{\left(\hbar \Gamma_{eg}\right)}^{2}}\frac{1}{i\left(\hbar \omega_{be}-\hbar \omega \right)+\hbar \Gamma_{be}}\;\;\;;\;\;\omega =\omega_{be} \end{array}$$

Analogous to the previous case, the frequency dispersion around two-photon resonance at $2\omega =\omega_{bg}$ can be presented as:(14)$$\begin{array}{l} \frac{i{\left|\mu_{eg}\right|}^{2}{\left|\mu_{be}\right|}^{2}}{4}\left[\frac{1}{{\left(E_{bind}\left(XX\right)/2\right)}^{2}+{\left(\hbar \Gamma_{eg}\right)}^{2}}-\frac{1}{{\left(iE_{bind}\left(XX\right)/2+\hbar \Gamma_{eg}\right)}^{2}}\right]\;\;\times \frac{1}{i\left(\hbar \omega_{bg}-2\hbar \omega \right)+\hbar \Gamma_{bg}}≃\\ \;\;\;\;\;\;\;\;\;\;\;\;\;\;\;\;\;\;\;\;\;\;\;\;\;\;\;\;\;≃\frac{i2{\left|\mu_{eg}\right|}^{2}{\left|\mu_{be}\right|}^{2}}{E_{bind}^{2}\left(XX\right)}\frac{1}{i\left(\hbar \omega_{bg}-2\hbar \omega \right)+\hbar \Gamma_{bg}} \end{array}$$
where $E_{bind}\left(XX\right)$ is the biexciton binding energy. The biexciton binding energy is defined as:(15)$$E_{bind}\left(XX\right)=2E\left(X\right)-E\left(XX\right),$$
where E(X) is the exciton energy.

After calculation of non-linear third-order susceptibility, it is possible to calculate linear, non-linear, and total absorption coefficients. The total absorption coefficient is a sum of two absorption coefficients, namely, linear and non-linear:(16)$$\alpha_{total}\left(\omega \right)=\alpha_{0}\left(\omega \right)+\alpha_{2}\left(\omega \right)I\left(\omega \right)$$

Here, $\alpha_{0}\left(\omega \right)=\frac{\hbar \Gamma_{eg}}{{\left(\hbar \omega -\hbar \omega_{eg}\right)}^{2}+{\left(\hbar \Gamma_{eg}\right)}^{2}}$, α2(ω)=32π2ωε0c2Imχ(3)(ω), and I(ω) is intensity of the incident light.

It is well known that biexciton states experimentally can be explored by observing two-photon absorption. As a consequence, we will calculate two-photon absorption in ellipsoidal QD. Two-photon absorption is connected with third-order susceptibility by:(17)α(ω)=4πωc(ε0)1/2I(ω)Imχ(3)

Around two-photon resonance, namely, $2\omega ≃\omega_{bg}$, Equation (14) will be transformed in:(18)$$\chi^{\left(3\right)}≅\frac{i2{\left|\mu_{eg}\right|}^{2}{\left|\mu_{be}\right|}^{2}}{E_{bind}^{2}\left(XX\right)\hbar \Gamma_{bg}}$$

## 4. Results and Discussion

After the detailed description of theory and formulation in Section 3, we will proceed to the discussion of the obtained results. Results are obtained for the *GaAs* QD with typical material parameters $m_{e}^{*}=0.067m_{0}$, $m_{h}^{*}=0.45m_{0}$, and ε=12.91, where $m_{0}$ is free electron mass and ε is the dielectric constant of the material [38]. As a next step, it is necessary to discuss biexciton energy levels for the ground and excited levels. We have considered levels, excited by holes and electrons. The results are presented in Figure 1.

In the energy diagrams we have used the following notations: $|ij\rangle \to e^{i}h^{j}\;$ for exciton and $|ijkl\rangle \to e^{i}e^{j}h^{k}h^{l}$ for biexciton, where i,j,k,l={0,1,2,…} is the particle level number. The first excited state of the biexciton is conditioned by the excited hole, while the state, corresponding to the excited state of electron, is higher. This is due to the difference between effective masses of the electron and hole. It is interesting that the state, corresponding to the second excited state of the hole, is close to the state, corresponding to the two first excited states of two holes (see first circle inset in the figure), creating a doublet state. By the same analogy, the triplet state can be found with merged states of |1100〉, |1020〉, and |2000〉.

Figure 2 shows the energy levels diagram, with corresponding transitions from biexciton states to exciton states and from exciton states to ground state. The diagram is plotted for the geometrical parameters of SOEQD c=5 nm and a=50 nm. We have considered only first three levels of the exciton and biexciton systems. The scheme of these transitions with their corresponding energies was used afterwards in order to obtain optical phenomena around one and two photon resonances.

The transitions, which are used in further calculations, are enumerated in Figure 2. Figure 3 shows the dependences of real and imaginary parts of the non-linear third-order susceptibility of the biexciton on the photon energy around one-photon resonance $\omega =\omega_{eg}$, around one-photon resonance $\omega =\omega_{be}$, and around two-photon resonance $\omega =\omega_{bg}$, as well as the total susceptibility of the photon energy for the fixed values of the ellipsoidal QD’s semiaxes.

It follows from the figure that for all cases, be it ground or excited level, the real parts of $\chi^{\left(3\right)}$ around one-photon resonance at $\omega =\omega_{be}$ and around two-photon resonance at $\omega =\omega_{bg}$ change their sign from positive to negative, while the imaginary parts of $\chi^{\left(3\right)}$ always have positive values. Contrary to this, the real part of $\chi^{\left(3\right)}$ around one-photon resonance at $\omega =\omega_{eg}$ changes its sign from negative to positive, and the imaginary part of $\chi^{\left(3\right)}$ is always negative. It is clear from the Figure 3a that the minimum and maximum values of real and imaginary parts of susceptibilities for transitions |00〉 and |01〉 are close to each other, while the values of real and imaginary parts of susceptibility for the transition |10〉 are weakened, which is explained by the weakness of the oscillator strength for this case. Contrary to the case around one-photon resonance at $\omega =\omega_{eg}$, the imaginary parts of the susceptibilities have two peaks around one-photon resonance at $\omega =\omega_{be}$. It is worth highlighting that for the ground level peaks positions are close to each other ($\hbar \omega_{eg}=56.5\;meV$ and $\hbar \omega_{be}=61.3\;meV$), which is why one can see one merged peak with high intensity in Figure 3b. On the other hand, for the excited levels the differences between transition energies are growing; thus, one can see clearly separated peaks in Figure 3b. Moreover, the maximum values of peaks for the imaginary parts of susceptibilities around $\omega =\omega_{be}$ and $\omega =\omega_{bg}$ are alongside each other. Finally, the real and imaginary parts of the total susceptibilities are presented in Figure 3d, taking into account the intensities’ ratio for one- and two-photon absorptions, which is of about four orders [11]. Note that the main contribution in total susceptibility is from one-photon resonances rather than two-photon resonance. However, for the absorption calculation the two-photon resonance will be important as the intense optical excitation will be applied and the probability of two-photon absorption will be increased.

In Figure 4, the same dependences as for Figure 3 have been plotted for the ground-to-ground transitions for different values of small semiaxis. As is obvious from Figure 4, the highest peak value has the curve for the highest value of the small semiaxis. The peak positions have red shift with the increase of the small semiaxis. This statement is valid both for the real and imaginary parts of the susceptibility. Here, we have investigated the impact of the small semiaxis since the size quantization in the axial direction is much more visual than in the radial direction. So, for all cases we kept the value of the large semiaxis as fixed.

Figure 5 and Figure 6 are devoted to the investigation of the absorption coefficient for the ground and excited levels, for the different values of the ellipsoidal QD’s geometrical parameters, respectively.

It can be seen from the Figure 5a that biexciton absorptions for ground and excited levels have the same peaks; however, the absorption for transition |10〉 is shifted to the higher energy. The same behavior can be found also in Figure 5b–d. To continue, if one looks over Figure 5b,d, one can conclude that the absorption curve around one-photon resonance at $\omega =\omega_{be}$ and total absorption curve for the ground level have one peak, while absorption curves for transitions |01〉 and |10〉 have two peaks. The origin of these peaks can be explained by the behavior of $\chi^{\left(3\right)}$ around one-photon resonance at $\omega =\omega_{be}$. Moreover, the total absorption for transition |10〉 have multiple peaks, which can be explained by the fact that the differences between energy levels play a part in the total energy, which, in its turn, result in the existence of many peaks. Finally, Figure 6 allows one to investigate the absorption and total absorption around one- and two-photon resonances for the different values of the ellipsoidal QD’s small geometrical parameter, when the large geometrical parameter is fixed.

Here, it is important to note that the highest peak has the absorption curve, calculated for the highest value of the small semiaxis of the ellipsoid, and the lowest peak has the curve for the lowest small semiaxis. Thus, the higher the small semiaxis’s value, the higher peak of the curve. The same approach is valid for the total absorption.

## 5. Conclusions

We have calculated the energies of biexciton states for ground and excited levels in GaAs ellipsoidal QD in the scope of the variational method. The trial variational functions have been constructed based on a single-particle wave function, having three variational parameters for the biexciton and one for the exciton. As the biexciton excited energy states have not been intensively studied in the literature, the energy diagram for the biexciton has been calculated up to 12th excited level for two sets of geometrical parameters. The shifts of the energy levels, conditioned by the change of the QD’s sizes, have been seen. For simplicity, we have considered only first three energy levels of the exciton and biexciton in order to construct the quantum transitions between these quasiparticles. Corresponding oscillator strengths of excitonic transitions, namely, for biexciton-exciton and exciton-ground state transitions, have been considered.

Instead of total third-order nonlinear optical susceptibility, obtained in the case of three-level model, the real and imaginary parts of third-order susceptibilities around one-photon resonances at $\omega =\omega_{eg}$ and $\omega =\omega_{be}$ for ground and excited levels, and around two-photon resonance at $\omega =\omega_{bg}$, have been discussed. It has been shown that for ground and excited levels, the real parts of $\chi^{\left(3\right)}$ around one-photon resonance at $\omega =\omega_{be}$ and around two-photon resonance at $\omega =\omega_{bg}$ change their sign from positive to negative, while the imaginary parts of $\chi^{\left(3\right)}$ always have positive values. Contrary to this, the real part of $\chi^{\left(3\right)}$ around one-photon resonance at $\omega =\omega_{eg}$ changes its sign from negative to positive, and the imaginary part of $\chi^{\left(3\right)}$ is always negative. It turns out that the strong peak for one-photon resonance at $\omega =\omega_{be}$ for the ground state consists of two merged peaks, while for the excited states these peaks are visibly separated. The ratio of the one-photon resonances to the two-photon resonance is about four orders in *GaAs* ellipsoidal QD. That is why the main contribution in the total susceptibility belongs to one-photon resonances. On the other hand, the contribution in the absorption spectra of two-photon resonance is growing because the incident light has high intensity and non-linear effects become important. Finally, the formula of the third-order susceptibility for the two-photon resonance has been obtained.

## Figures and Tables

**Figure 1 nanomaterials-12-01412-f001:**
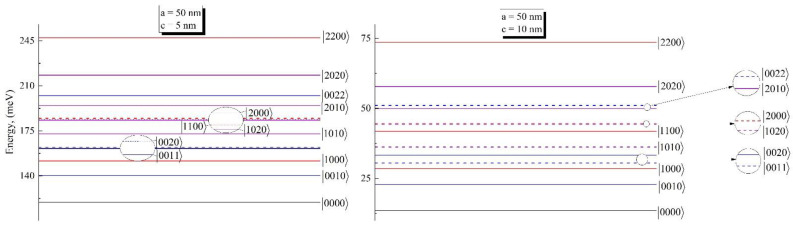
The energy diagram for the biexciton of ground and excited states for different sets of geometrical parameters.

**Figure 2 nanomaterials-12-01412-f002:**
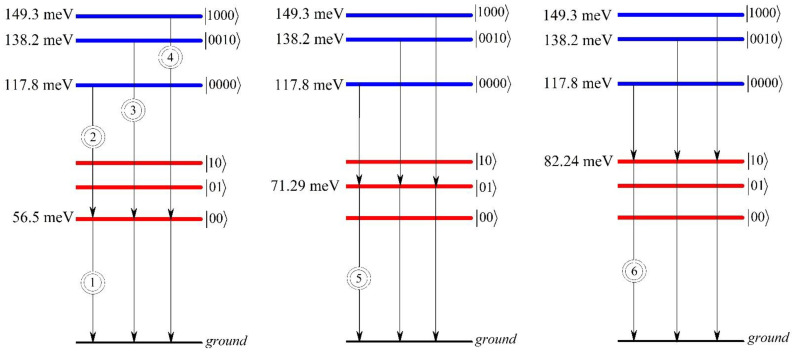
The transition diagram for the biexciton, exciton, and ground states. The geometrical parameters of the ellipsoidal QD have been chosen as follows: c=5 nm and a=50 nm.

**Figure 3 nanomaterials-12-01412-f003:**
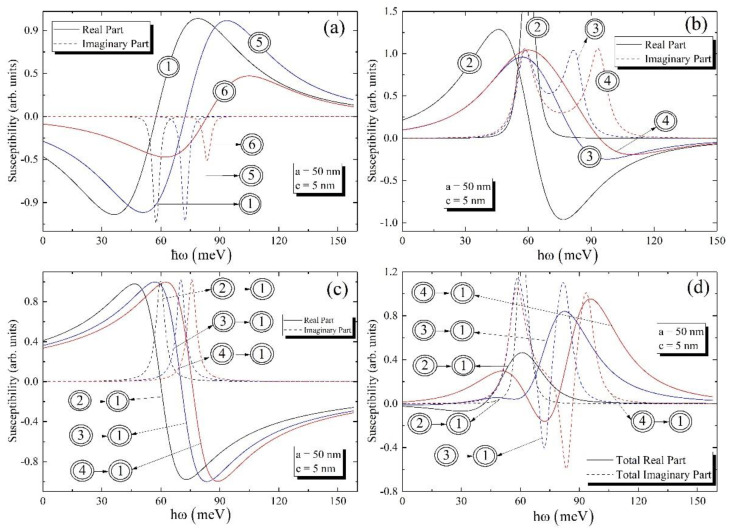
The real and imaginary parts of the susceptibility of ground and excited levels (**a**) around one-photon resonance at $\omega =\omega_{eg}$, (**b**) around one-photon resonance at $\omega =\omega_{be}$, and (**c**) around two-photon resonance at $\omega =\omega_{bg}$; (**d**) the dependence of the real and imaginary parts of the total susceptibility on the photon energy.

**Figure 4 nanomaterials-12-01412-f004:**
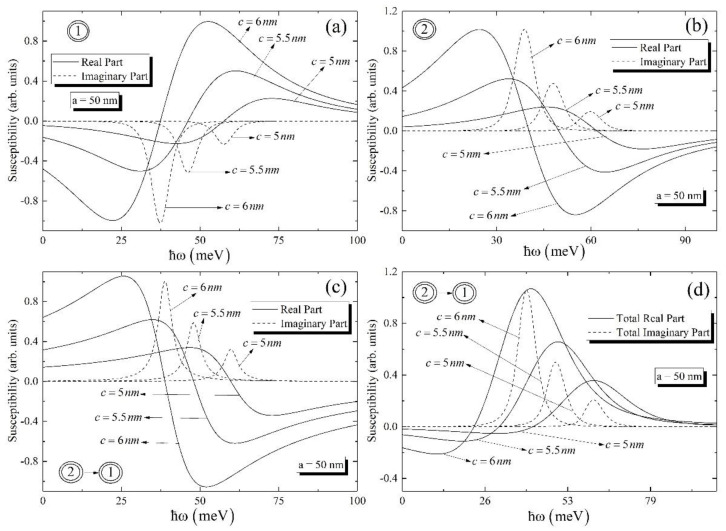
The real and imaginary parts of the susceptibility for ground levels for different small semiaxis (**a**) around one-photon resonance at $\omega =\omega_{eg}$, (**b**) around one-photon resonance at $\omega =\omega_{be}$, and (**c**) around two-photon resonance at $\omega =\omega_{bg}$; (**d**) the dependence of the real and imaginary parts of the total susceptibility on the photon energy.

**Figure 5 nanomaterials-12-01412-f005:**
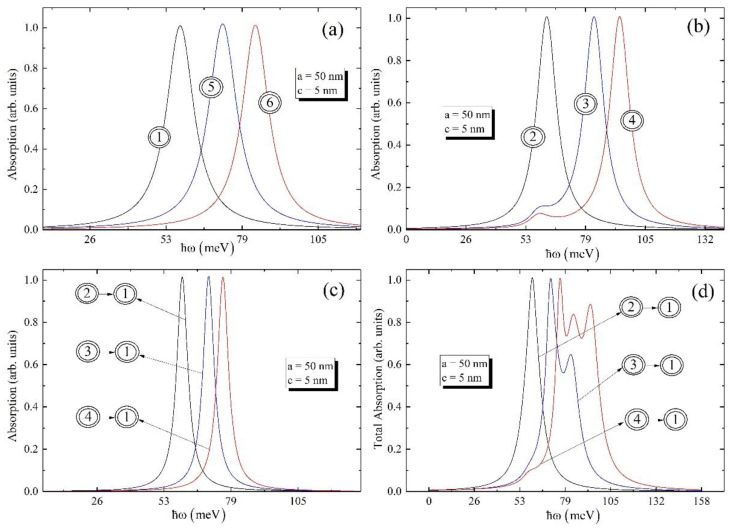
The absorption of ground and excited levels (**a**) around one-photon resonance at $\omega =\omega_{eg}$, (**b**) around one-photon resonance at, $\omega =\omega_{be}$, and (**c**) around two-photon resonance at $\omega =\omega_{bg}$; (**d**) the dependence of the total absorption on the photon energy.

**Figure 6 nanomaterials-12-01412-f006:**
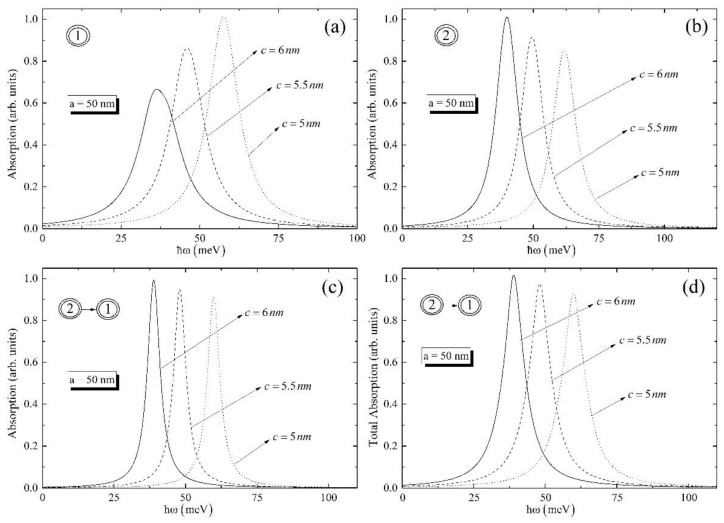
The absorption coefficient for ground levels for different small semiaxis (**a**) around one-photon resonance at $\omega =\omega_{eg}$, (**b**) around one-photon resonance at, $\omega =\omega_{be}$, and (**c**) around two-photon resonance at $\omega =\omega_{bg}$; (**d**) the dependence of the total absorption on the photon energy.

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
