# Peer review of "Non-Linear Optical Properties of Biexciton in Ellipsoidal Quantum Dot"

_nanomaterials, 2022, doi:10.3390/nano12091412_

Round 1
Reviewer 1 Report
In this work the authors reported the theoretical investigation on the non-linear optical properties induced by excition and biexcition in ellipsoidal quantum dot. Considering their previously reported study on the energy levels and direct interband light absorption in a strongly oblated ellipsoidal quantum dot based on the framework of adiabatic approximation, and the detailed derivative formulae for the transition, the results are believed to be correct. Overally this work is interesting and I think it can be accepted for publication by addressing the following comments,
1) In the calculations of optical properties using the formulae they derived, parameters of GaAs were used. I think the discussion on the comparison between model prediction and real experiments should be provided.
2) If the boundary condition is changed to allow the leak of energy, what will happen?
3) The typos should be corrected.
Author Response
Dear Editor,
I am sending you the revised version of the manuscript entitled “Non-Linear Optical Properties of Biexciton in Ellipsoidal Quantum Dot” taking into consideration the reviewers’ comments and recommendations. The authors are grateful to the reviewers for careful and detailed reading of the manuscript and useful remarks and recommendations. The manuscript has been revised taking into account all suggestions of the reviewers.
Comments for Reviewer 1
- In the calculations of optical properties using the formulae they derived, parameters of GaAs were used. I think the discussion on the comparison between model prediction and real experiments should be provided.
Unfortunately, there are no experimental works related to the biexciton states in GaAs ellipsoidal quantum dot. At least we didn’t find such works. We hope that such investigation will be implemented in the future.
- If the boundary condition is changed to allow the leak of energy, what will happen?
In the case, when the finite confinement potential is considered, some new physical phenomena will appear. For example, the possibility of the leakage of the electron and hole for the quantum dot to the surrounding media. It is obvious, that in this case the biexciton binding energy will be less and the biexciton stability can be broken, depending on the geometrical parameters of the ellipsoidal quantum dot. However, we have considered the case, when the quantum dot is surrounded by the dielectric media, so that the confinement energy prevails over the energies, mentioned in the article. From the theoretical point of view, the problem becomes more complicated and requires special numerical methods for the investigation. Note, that this part has been added to the article.
- The typos should be corrected.
According to the reviewer comments, all minor typos have been corrected.
Reviewer 2 Report
The issue of the studying optical properties of biexcitons in quantum dots has nowadays attracted very considerable attention. This is one of the most urgent problems of modern nanophysics, semiconductor physics, etc. The paper by Bleyan et al is devoted to studying non-linear optical properties of biexciton in еру GaAs ellipsoidal quantum dot. The authors have presented the results of a theoretical study of exciton and biexciton states for a group of ground and excited levels in the GaAs strongly oblate ellipsoidal quantum dot. The standard variational principle of quantum mechanics has been applied. The authors have used the variational trial wave functions for the ground and excited states of the exciton and biexciton, which are “constructed on the basis of 18 single-particle wave functions”. The authors have calculated the oscillator strengths for transitions from the excitonic to the biexciton states as the functions of the minor and major semiaxes of an ellipsoidal quantum dot. The coefficient of two-photon absorption of a biexciton in an ellipsoidal quantum dot is calculated. The title is quite adequate and appropriate. The introduction is a very useful tour d'horizon to state of art of the optical and spectral properties of excitons and biexcitons. The abstract contains the essential information of the article. All theoretical aspects and calculations are carried out very carefully and do not contain any significant misprints. The parts describing the main results are clearly and correctly written.
The paper is very actual, interesting, original and novel and adds so much to results that are already published. My understanding is that the paper by Bleyan et al is aimed for the MDPI journal “Nanomaterials” so its style is adequate to the purpose, and the length of a paper is justified by its contents.
However, there are a few minor points, which should be clarified in order to meet the possible questions of the readers and to make the contribution even more appealing to a wider audience (with the consent of the authors):
i). The abstract as well as the conclusions of the article, should be corrected. Namely, it would be worth specifying both in the abstract and in the conclusions that all results are obtained for the GaAs quantum dot (QD). The authors should clearly explain all the abbreviations used.
ii) Very important, novel results are presented by the authors in Section 3, which is devoted to the study of optical properties. In particular, the authors determined the strength of the oscillator of the transition from the exciton to the biexciton state and oscillator strengths of the exciton transitions. The authors have also analyzed the real and imaginary parts of the third-order susceptibilities around one-photon resonances for the ground and excited levels, and around two-photon resonances. The trial variational functions have been constructed based on the single-particle wave functions, having three variational parameters for the biexciton and one for the exciton. The standard Fermi rule is used to calculate the oscillator strengths in the “velocity” form etc. It would be extremely interesting and useful for the reader to understand how consistently and correctly the wave functions of the studied states are constructed. In addition, it is important to verify compliance with the principle of gauge invariance when calculating oscillator strengths and, further, susceptibilities. In particular, how much would the oscillator strengths calculated using the velocity and length formulas differ (depending on the choice of the shape of the photon propagator). The short comment is very desirable.
iii). With the use of high-power ultrashort laser pulses, a lot of data have appeared on nonlinear optical processes in semiconductors and semiconductor nanostructures. Among the studies of the nonlinear optical properties of semiconductor nanostructures, it is useful to note the papers on two-photon absorption, second-harmonic generation, as well as a gauge invariance in the radiation amplitudes etc. Some additional references can be very useful for readers (at the author’s discretion):
- Ivchenko E.I., Pikus P.E., Superlattices and Other Heterostructures: Symmetry and Optical Phenomena / Berlin: Springer, 1995. 386 p.
- Fedorov A.V., Baranov A.V., Inoue K., Two-photon transitions in systems with semiconductor quantum dotsю Phys. Rev. B. 1996. V. 54. P. 8627-8632
- Norris D., Sacra A., Murray C., Bawendi M., Measurement of the size dependent hole spectrum in CdSe quantum dots. Phys. Rev. Lett. 1994. V.72. P. 2612-2615
- Wang K.L., Balandin A.A., Quantum Dots: Physics and Applications, in Optics of Nanostructured Materials, edited by V. Markel and T. George / New York: John Wiley & Sons, Inc.,2001. 532 p.
- Baranov A.V., Inoue K., Toba K., Yamanaka A., Petrov V.I., Fedorov A.V., Resonant hyper-Raman and second-harmonic scattering in a CdS quantum-dot system. Phys. Rev. B. 1996.V.53. P. 1721
- Glushkov A.V., Ivanov L.N., Radiation decay of atomic states: atomic residue polarization and gauge noninvariant contributions. Phys.Lett.A. 1992. Vol.170, N1. P.33-36.
- Glushkov A.V., Multiphoton spectroscopy of atoms and nuclei in a laser field: Relativistic energy approach and radiation atomic lines moments method. Adv Quant Chem. (Elsevier) 2019, V.78, P. 253-285;
- Dneprovskii V., Kozlova M., Smirnov A., Wumaier T., The features of nonlinear absorption of two-photon excited excitons in CdSe/ZnS quantum dots. Phys. E. 2012. V. 44. P.1920–1923.
- Pokutnyi S.I., Jacak L., Misiewicz J., Salejda W., Zegrya G.G., Stark effect in semiconductor quantum dots J. Appl. Phys. 2004. V. 96(2). P. 1115-1119
v) Please, the authors should carefully proof-read the manuscript to minimize typographical, editorial or other misprints (e.g. page 1, lime 39: “ Shtark effect [8]” etc), check that all symbols (in the text, formulas, figures) and abbreviators are defined etc.
Recommendation: The scientific merit of the paper is very high. The paper should be recommended for publication in the MDPI journal “Nanomaterials” provided the authors comply with a few minor points listed. Thanks to authors for minor revision.
Author Response
Dear Editor,
I am sending you the revised version of the manuscript entitled “Non-Linear Optical Properties of Biexciton in Ellipsoidal Quantum Dot” taking into consideration the reviewers’ comments and recommendations. The authors are grateful to the reviewers for careful and detailed reading of the manuscript and useful remarks and recommendations. The manuscript has been revised taking into account all suggestions of the reviewers.
Comments for Reviewer 2
- The abstract as well as the conclusions of the article, should be corrected. Namely, it would be worth specifying both in the abstract and in the conclusions that all results are obtained for the GaAs quantum dot (QD). The authors should clearly explain all the abbreviations used.
According to the reviewer comment, the abstract as well as conclusion have been updated, and all abbreviations are clearly explained.
- Very important, novel results are presented by the authors in Section 3, which is devoted to the study of optical properties. In particular, the authors determined the strength of the oscillator of the transition from the exciton to the biexciton state and oscillator strengths of the exciton transitions. The authors have also analyzed the real and imaginary parts of the third-order susceptibilities around one-photon resonances for the ground and excited levels, and around two-photon resonances. The trial variational functions have been constructed based on the single-particle wave functions, having three variational parameters for the biexciton and one for the exciton. The standard Fermi rule is used to calculate the oscillator strengths in the “velocity” form etc. It would be extremely interesting and useful for the reader to understand how consistently and correctly the wave functions of the studied states are constructed. In addition, it is important to verify compliance with the principle of gauge invariance when calculating oscillator strengths and, further, susceptibilities. In particular, how much would the oscillator strengths calculated using the velocity and length formulas differ (depending on the choice of the shape of the photon propagator). The short comment is very desirable.
The system, considered in this paper, is described in the framework of the nonrelativistic approximation and in the absence of external fields. Therefore, the issue of gauge arises only when the operator of the incident perturbation is chosen. The corresponding discussion has been added to the manuscript.
- With the use of high-power ultrashort laser pulses, a lot of data have appeared on nonlinear optical processes in semiconductors and semiconductor nanostructures. Among the studies of the nonlinear optical properties of semiconductor nanostructures, it is useful to note the papers on two-photon absorption, second-harmonic generation, as well as a gauge invariance in the radiation amplitudes etc. Some additional references can be very useful for readers (at the author’s discretion):
- Ivchenko E.I., Pikus P.E., Superlattices and Other Heterostructures: Symmetry and Optical Phenomena / Berlin: Springer, 1995. 386 p.
- Fedorov A.V., Baranov A.V., Inoue K., Two-photon transitions in systems with semiconductor quantum dotsю Phys. Rev. B. 1996. V. 54. P. 8627-8632
- Norris D., Sacra A., Murray C., Bawendi M., Measurement of the size dependent hole spectrum in CdSe quantum dots. Phys. Rev. Lett. 1994. V.72. P. 2612-2615
- Wang K.L., Balandin A.A., Quantum Dots: Physics and Applications, in Optics of Nanostructured Materials, edited by V. Markel and T. George / New York: John Wiley & Sons, Inc.,2001. 532 p.
- Baranov A.V., Inoue K., Toba K., Yamanaka A., Petrov V.I., Fedorov A.V., Resonant hyper-Raman and second-harmonic scattering in a CdS quantum-dot system. Phys. Rev. B. 1996.V.53. P. 1721
- Glushkov A.V., Ivanov L.N., Radiation decay of atomic states: atomic residue polarization and gauge noninvariant contributions. Phys.Lett.A. 1992. Vol.170, N1. P.33-36.
- Glushkov A.V., Multiphoton spectroscopy of atoms and nuclei in a laser field: Relativistic energy approach and radiation atomic lines moments method. Adv Quant Chem. (Elsevier) 2019, V.78, P. 253-285;
- Dneprovskii V., Kozlova M., Smirnov A., Wumaier T., The features of nonlinear absorption of two-photon excited excitons in CdSe/ZnS quantum dots. Phys. E. 2012. V. 44. P.1920–1923.
- Pokutnyi S.I., Jacak L., Misiewicz J., Salejda W., Zegrya G.G., Stark effect in semiconductor q uantum dots J. Appl. Phys. 2004. V. 96(2). P. 1115-1119
Thank you for provided references. Some of them have been discussed and added to the manuscript.
- Please, the authors should carefully proof-read the manuscript to minimize typographical, editorial or other misprints (e.g. page 1, lime 39: “ Shtark effect [8]” etc), check that all symbols (in the text, formulas, figures) and abbreviators are defined etc.
According to the reviewer comment, the manuscript has been updated.
Round 2
Reviewer 1 Report
The authors have addressed all my concerns in the revised manuscript.